# Dietary Oyster (*Crassostrea gigas*) Extract Ameliorates Dextran Sulfate Sodium-Induced Chronic Experimental Colitis by Improving the Composition of Gut Microbiota in Mice

**DOI:** 10.3390/foods11142032

**Published:** 2022-07-08

**Authors:** Tatsuya Ishida, Hiroyuki Matsui, Yoshikazu Matsuda, Takaki Shimono, Seiji Kanda, Toshimasa Nishiyama, Ryota Hosomi, Kenji Fukunaga, Munehiro Yoshida

**Affiliations:** 1Central Research Institute, Japan Clinic Co., Ltd., 1 Nishimachi, Taishogun, Kyoto 603-8331, Japan; tatsuya.ishida@japanclinic.co.jp (T.I.); hiroyuki.matsui@japanclinic.co.jp (H.M.); yoshikazu.matsuda@japanclinic.co.jp (Y.M.); 2Department of Hygiene and Public Health, Kansai Medical University, 2-5-1 Shin-machi, Osaka 573-1010, Japan; shimonot@hirakata.kmu.ac.jp (T.S.); kandas@hirakata.kmu.ac.jp (S.K.); tnishi@takii.kmu.ac.jp (T.N.); 3Faculty of Chemistry, Materials, and Bioengineering, Kansai University, 3-3-35 Yamate-cho, Osaka 564-8680, Japan; fukunagk@kansai-u.ac.jp (K.F.); hanmyou4@kansai-u.ac.jp (M.Y.)

**Keywords:** oyster extract, Pacific oysters, *Crassostrea gigas*, chronic experimental colitis, microbiota, short-chain fatty acid

## Abstract

Previously, we have reported that the intake of oyster extract (OE), prepared from Pacific oysters (*Crassostrea gigas*), can attenuate symptoms of dextran sulfate sodium (DSS)-induced acute experimental colitis in mice. Herein, we aimed to evaluate whether OE intake ameliorates chronic experimental colitis induced by repeated DSS administration in mice. Male C57BL/6J (4-week-old) mice were fed either the standard diet AIN93G (control diet) or the control diet containing 5.0% (*w*/*w*) OE (OE diet). After 21 days of diet feeding, chronic experimental colitis was induced by three cycles of 2.0% (*w*/*w*) DSS solution administration (5 days), followed by distilled water (5 days). Mice fed OE alleviated the shortened colonic length, increased the relative weight of the spleen, colonic histopathological score (regeneration), and blood in the stool score compared with mice fed control diet. A tendency to improve the α-diversity of fecal microbiota, which was exacerbated by colitis, was observed in mice fed OE. Correlation analysis suggested that the anti-colitis effect of OE intake could be related to the valeric acid content and relative abundances of *Ruminococcus* and *Enterococcus* in the feces. In conclusion, OE could ameliorate DSS-induced chronic experimental colitis by improving the gut environment, including the microbiota community and SCFA composition.

## 1. Introduction

Oysters are among the most widely distributed marine species worldwide. For example, Pacific oysters (*Crassostrea gigas*) and rock oysters (*Crassostrea nippona*) can be found in Japan. Oysters are consumed in several countries, either raw, grilled, broiled, steamed, or fried. Oysters have long been called the “milk of the sea”, owing to their rich content of polysaccharides, proteins, vitamins, and minerals [1]. Importantly, oysters have high potential as functional foods, given their high nutritional value and health benefits. Health-promoting components in oysters include polysaccharides, taurine, polypeptides, and polyphenols, which can be concentrated using various extraction methods [2]. Previous studies have reported that the intake of oyster extract (OE) can afford numerous health benefits, such as antioxidant [3], antimicrobial [4], antitumor [5], antiplatelet aggregation [6], and antihyperglycemic effects [7]. We have previously reported the health-promoting functions of OE prepared from *C. gigas*, mainly composed of polysaccharides, taurine, proteins, amino acids, and zinc, which inhibits the initiator action of carcinogens [8], reduces the hepatic cholesterol content [9], and accelerates the recovery of cell function in proximal tubular epithelial cells following nephrotoxicity [10]. Therefore, OE prepared from *C. gigas* has received considerable attention as a potential functional food.

The gut environment, including the microbiota community and bacterial metabolites, is potently influenced by host conditions and, in turn, impacts the host metabolism, immunity, and endocrine functions [11]. The gut environment is influenced by food components, particularly fiber, resistant starch, and resistant protein [12]. In contrast, we have reported that dietary OE can affect the gut microbiota community and bacterial metabolites such as short-chain fatty acids (SCFA) in rodent models [13,14]. On examining the effect of OE intake on inflammatory bowel disease (IBD), a well-known underlying cause of gut environment deterioration, OE, could alleviate symptoms of acute experimental colitis in a dextran sulfate sodium (DSS)-induced mouse model, partly by improving the gut environment, including the microbiota community (composition and diversity) and SCFA composition [15]. However, human IBD is a chronic symptom of this relapsing and remitting state, and an acute experimental colitis mouse model does not comprehensively simulate human IBD. Accordingly, a more appropriate method has been developed to establish chronic experimental colitis in mice by administering DSS repeatedly, which would more closely mimic human IBD symptoms [16,17]. Therefore, it would be beneficial to evaluate OE in both chronic and acute experimental colitis models to comprehensively clarify the anti-colitis effects. Herein, we examined the effects of OE prepared from *C. gigas* intake on colitis symptoms, fecal microbiota, and SCFA composition in a chronic experimental colitis mouse model induced by repeated DSS administration.

## 2. Materials and Methods

### 2.1. Preparation and Nutrients Composition of OE

The preparation of OE from *C. gigas*, as well as its nutrient composition, have been described in our previous report [15].

### 2.2. Approval for Animal Experiments

The experimental animal protocol was reviewed and approved by the Animal Ethics Committee of Kansai University (approval no. 1918, 11 May 2019). The humane endpoint was set as body weight (BW) reduction of 20% or more when compared with BW at initiating DSS administration.

### 2.3. Animal Diet and Care

Appendix A presents the composition of the experimental diet based on AIN-93G [18], with equal amounts of carbohydrate (645 g/kg), protein (200 g/kg), fat (70 g/kg), and sodium chloride (4.12 g/kg). As reported in our previous studies [9,14,15], the OE concentration in the experimental diet was set at 5.0% (*w*/*w*). Mice were housed in individual breeding cages (cat. KN-60105-T; Natsume Seisakusho Co., Ltd., Tokyo, Japan) and maintained in air-conditioned rooms with free access to water and the experimental diets.

### 2.4. Animal Experiment

Herein, we employed male C57BL/6J mice (4-week-old) purchased from Japan SLC, Inc. (Hamamatsu, Japan). Mice were brought to the breeding room at Kansai University and then acclimated for seven days while receiving the control diet. Experimental mice were divided into three groups with similar mean BW and standard deviation (control [*n* = 6], control + DSS [*n* = 8], and OE + DSS [*n* = 8] groups). The control and control + DSS groups were fed the control diet, whereas the OE + DSS group received the OE diet. Food and water intake and BW were measured every two days.

Chronic experimental colitis was induced according to the schedule presented in Appendix A, based on a previous report [17]. After three weeks of the experimental diet feeding, DSS was administered to induce chronic experimental colitis. Mice were administered a 2.0% (*w*/*w*) DSS (MP Biomedicals, Irvine, CA, USA) solution for 5 days, followed by distilled water for 5 days (washout period); this cycle was repeated three times. The disease activity index (DAI) scores [19] were assessed every two days at 10:00 AM for a 30-day period, from day 0. No mice warranted the application of humane endpoints.

On day 30, feces remaining in each mouse cage were collected. Under non-fasting conditions, blood collection (9:00–12:00) was performed in isoflurane-anesthetized mice, and euthanasia was induced by excessive isoflurane inhalation. Serum was obtained by centrifugation (2000× *g* for 15 min). The spleen, liver, kidneys, perirenal white adipose tissue, small intestine, cecum, and colon were excised and then weighed. The colon length was also measured. The distal colon was washed with cold saline to remove any colon contents and stored in Gene Keeper RNA & DNA stabilization solution (cat. 319-08901; Nippon Gene Co., Ltd., Tokyo, Japan) and 10% formalin solution for Tissue Fixation (Fujifilm Wako Pure Chemicals, Osaka, Japan), respectively. Hematoxylin and eosin (HE)-stained distal colon specimens were prepared and scored for inflammation, crypt damage, regeneration, and extent, as previously reported [15,20].

### 2.5. Analysis of Serum Biochemical Parameters

Serum biochemical parameters (total protein, albumin, aspartate aminotransferase, alanine aminotransferase, creatine phosphokinase, lactate dehydrogenase, urea nitrogen, creatinine, triglyceride, total cholesterol, high-density lipoprotein cholesterol, and phospholipids) were measured using a commercial service (Japan Medical Laboratory, Kaizuka, Japan).

### 2.6. Analysis of Gene Expression

Expression levels of tumor necrosis factor (*Tnf*) α, interleukin (*Il*) *1β*, and *Il6* in the mucosa of the distal colon were measured by quantitative PCR, as described in our previous report [21].

### 2.7. Analysis of Fecal SCFA Compositions

Fecal SCFA composition was measured using a gas chromatography-flame ionization detector (GC-2014, Shimadzu Co., Kyoto, Japan), as described in our previous report [22].

### 2.8. Analysis of 16S rRNA Amplicon Sequence

Six fecal samples from all animals in the control group, control + DSS, and OE + DSS groups, respectively, were randomly selected, and total DNA was extracted using ISOSPIN Fecal DNA (cat. 315-08621; Nippon Gene Co., Ltd.). The microbiota community (composition and diversity) were analyzed by 16S rRNA amplicon sequencing using a next-generation sequencer Ion PGM^TM^ workflow (Thermo Fisher Scientific Inc., Waltham, MA, USA) following our previous report [15]. β-Diversity and linear discriminant analysis (LDA) effect size (LEfSe) [23] were visualized using ClustVis (https://biit.cs.ut.ee/clustvis/ (accessed on 12 April 2022) and Galaxy (http://huttenhower.sph.harvard.edu/galaxy/ (accessed on 13 April 2022).

### 2.9. Statistical Analysis of Data

Data are presented as the mean values and standard errors of the mean. We examined statistically significant differences between the control and control + DSS groups and between the control + DSS and OE + DSS groups. Parametric and nonparametric data were subjected to one-way analysis of variance followed by the Holm–Sidak multiple comparison test and Kruskal–Wallis test followed by the uncorrected Dunn test for statistical difference, respectively. This study did not compare the control and OE + DSS group. This is because there are two factors (DSS administration and OE intake) between the groups, and comparing these two groups would not reveal the factor responsible for the difference. A *p*-value < 0.01, *p*-value < 0.05, and 0.05 ≤ *p*-value < 0.10 were considered statistically significant and statistical tendency, respectively. Data analyses were conducted using GraphPad Prism software (version 7.0d; GraphPad Software, San Diego, CA, USA) on an iMac (Mid 2014, Apple Inc., Cupertino, CA, USA).

## 3. Results

### 3.1. Growth Parameters, Change of BW, Colon Length, and Organs Weight

The examined groups exhibited no significant differences in food and water intake during the experimental period and DSS solution intake during the DSS administration period (Appendix A). Figure 1 presents the indicators of chronic experimental colitis severity (changes in BW, colon length, and relative weight of the spleen). Following DSS administration (the control + DSS and OE + DSS groups), BW decreased on day 6, subsequently increasing steadily. There was no change in BW between the DSS administration groups.

Shortened colon length and increased relative spleen weight were used as biological markers of DSS-induced colitis severity [24]. DSS administration altered these two markers in the direction of worsening colitis, and OE intake significantly alleviated these DSS-exacerbated markers (Figure 1C,D). In contrast, DSS administration influenced the relative weight of the liver, small intestine, cecum, and colon, but not that of the kidney and perirenal white adipose tissue (Appendix A). The relative organ weights did not significantly differ between the control + DSS and OE + DSS groups.

### 3.2. Change in DAI Scores

Figure 2 presents changes in BW loss, blood in the stool, stool consistency, and DAI scores. The DSS administration groups displayed increased blood in the stool, stool consistency, and DAI scores from day 2 (Figure 2B–D). In addition, the area under the curve (AUC) of BW loss, stool consistency, blood in the stool, and DAI scores were significantly higher in the control + DSS group than in the control group (Figure 2E–H). The OE + DSS group had a significantly lower blood in the stool score than the control + DSS group on day 16 (Figure 1B). However, the AUC of the OE + DSS group did not significantly differ from that of the control + DSS group (Figure 2E–H).

### 3.3. Serum Biochemical Parameters

No significant differences in serum biochemical parameters (total protein, albumin, alanine aminotransferase, creatine phosphokinase, lactate dehydrogenase, urea nitrogen, creatinine, triglyceride, phospholipids, total cholesterol, and high-density lipoprotein cholesterol) were detected between examined groups (Appendix A).

### 3.4. Histopathological Damage in the Colonic Tissue

Figure 3 presents the histopathological grading of colonic tissue specimens. DSS administration increased histopathological damage (inflammation, crypt damage, regeneration, and extent) in colonic tissues (Figure 3B–E). Compared with the control + DSS group, the OE + DSS group exhibited significantly reduced regeneration scores (*p* = 0.03, Figure 3D).

### 3.5. Gene Expression Levels in the Colonic Mucosa

Figure 4 presents the mRNA expression levels of *Tnfα*, *Il1β*, and *Il6*. DSS administration significantly increased *Tnfα* expression (Figure 4A). In contrast, expression levels of *Il1β*, *Il6*, and *Tnfα* in the OE + DSS group did not significantly differ from those in the control + DSS group.

### 3.6. Fecal SCFA Contents and Compositions

Figure 5 represents the fecal SCFA content and composition. DSS administration significantly increased the total SCFA and individual SCFA contents (acetic acid, propionic acid, isobutyric acid, butyric acid, and isovaleric acid) in the feces (Figure 5A,B), accompanied by a significant decrease in the relative fecal content of valeric acid (Figure 5C). The fecal acetic acid content tended to be lower in the OE + DSS group than in the control + DSS group (*p* = 0.07, Figure 5A).

### 3.7. Diversity and Composition of Fecal Microbiota

Figure 6 shows the structure and composition of fecal microbiota. After processing by Ion Reporter (Thermo Fisher Scientific Inc.), the number of total reads (the control group: 179,787 ± 18,479, the control + DSS group: 187,456 ± 20,677, and the OE + DSS group: 161,965 ± 26,451) and valid reads (the control group: 110,369 ± 17,423, the control + DSS group: 118,994 ± 17,998, and the OE + DSS group: 105,498 ± 20,893) did not significantly differ between examined groups. The Chao-1 index showed no significant differences between groups (Figure 6A). In contrast, OE intake restored the Simpson index, which was reduced by DSS administration (Figure 6B). Moreover, DSS administration altered the principal component analysis of bacterial communities (Figure 6C).

Histograms of the relative abundances of bacterial phyla and genera in fecal specimens are presented in Appendix A. At the phylum level, the control + DSS group showed an increased relative abundance of Proteobacteria when compared with the control group (Figure 6D), whereas OE intake significantly increased the relative abundance of Bacteroidetes when compared with the control +DSS group (Figure 6E). At the genus level, DSS administration decreased the relative abundances of *Bifidobacterium*, [*Ruminococcus*], and *Ruminococcus* (Figure 6H–J), and increased the relative abundance of *Enterococcus* (Figure 6K). Compared with the control + DSS group, the OE + DSS group showed a significantly increased relative abundance of *Ruminococcus* (Figure 6G), tended to increase the relative abundance of *Bacteroides* (Figure 6J), and tended to decrease the relative abundance of *Enterococcus* (Figure 6K).

Figure 7 presents the LDA histogram scores and the LEfSe cladogram. Among the examined groups, we identified 29 distinct taxa (from phylum to genus levels). Actinobacteria, Bacteroidetes, and Firmicutes phylotypes were markedly abundant in the control, control + DSS, and OE + DSS groups at the phylum level, respectively.

## 4. Discussion

To determine the potential of OE in treating symptoms of chronic experimental colitis, we examined the effects of OE on colitis symptoms, gut microbiota community, and SCFA in an experimental mouse model of chronic colitis induced by three repeated cycles of DSS administration. In the histopathological evaluation of the colon (Figure 3A), we observed mononuclear plasma cell infiltration [17] and locally thickened mucosa [16], characteristics of chronic experimental colitis mouse models. Repeated DSS administration phenotypically demonstrated that a chronic experimental colitis mouse model was successfully established in the present study. Compared with the control + DSS group, OE intake (the OE + DSS group) could alleviate colonic length shortening, as well as increase in the relative spleen weight, colonic histopathological scores (regeneration), and blood in the stool scores on day 16 (a DAI evaluation item). Previously, we reported that OE intake has an ameliorative effect on acute experimental colitis induced by DSS administration [15]. These indicate that OE intake ameliorates acute and chronic experimental colitis induced by DSS administration.

On the other hand, OE intake did not affect the expression of inflammation-related genes in colonic mucosa (Figure 4), one of the indicators of the severity of colitis symptoms. The mechanism by which DSS induces colitis is unknown but may be the result of damage to the epithelial monolayer lining the colon and diffusion of inflammatory gut contents (e.g., bacteria and their products) into the underlying tissue [25]. OE intake may not be effective in preventing the diffusion of inflammatory gut contents into the underlying tissues. On the other hand, relative spleen weight, an indirect indicator of the severity of colitis symptoms, was significantly reduced by OE intake (Figure 1D). Since considerable amounts of DSS penetrated from the gut is found in the spleen during the chronic phase of DSS-induced colitis [26], it is believed that the immune response elicited by DSS increases spleen weight. OE intake maintained gut barrier function and inhibited DSS penetration, which led to the suppression of spleen weight gain. To clarify the effect of OE intake on the maintenance of gut barrier function, it is necessary to examine the expression of gut tight junction proteins such as claudin, occludin, and zonula occluden, and the translocation of fluorescein isothiocyanate-labeled dextran into the blood. In addition, OE intake was associated with lower regeneration score in colonic tissue (Figure 3D). Growth factors, including intestinal epithelial cell-specific insulin-like growth factor 1 [27] and epidermal growth factor [28], are known to promote epithelial cell regeneration. Recent studies reported that the activation of transmembrane G protein-coupled receptor 5 (TGR5) by bile acids can promote regeneration of the intestinal epithelium, which helps restore the gut mucosal barrier following some disruption [29]. Taurine, which is also contained in OE, has been reported to alter the ileum bile acid composition [30], and this alteration may activate TGR5 and promote epithelial cell regeneration. To elucidate the mechanism by which OE intake improves gut epithelial cell regeneration, the effects on growth factor and TGR5 in the gut need to be examined.

Although the pathogenesis of IBD is complex, alterations in the gut microbiota community and metabolites are among the most crucial findings related to IBD pathogenesis [31,32]. In particular, we aimed to examine the effects of OE on fecal microbiota and SCFA in a chronic experimental colitis mouse model. Major gut SCFA, including acetic acid, propionic acid, and butyric acid, are produced by gut bacteria-mediated fermentation of indigestible polysaccharides and proteins, and they play a crucial role in maintaining the intestinal environment [33]. In a previous study, the SCFA content in the colon and feces of mice with colitis-induced by DSS was found to be reduced [34], whereas the fecal content of major SCFA and their total content in the control + DSS group were elevated when compared with the control group (Figure 5A,B). Previous reports have shown that patients with IBD exhibit decreased colonic SCFA absorption and metabolism [35,36]. Moreover, acetic acid has been used to induce colitis in a mouse model [37]. A strong correlation has been noted between indicators of chronic experimental colitis severity and the fecal SCFA content (Appendix A). We presented similar results in our previous study [15]. Therefore, the elevated fecal acetic acid and total SCFA contents in mice with chronic experimental colitis may be related to colon dysfunction. Compared with the control + DSS group, the fecal acetic acid and total SCFA content tended to decrease (Figure 5A,B). The OE-induced decrease in fecal SCFA content indicates a reduction in chronic experimental colitis severity, potentially attributed to colonic SCFA absorption by reducing the regeneration score (Figure 3D) and extension of colon length (Figure 1C).

Compared with the control group, the control + DSS group showed a decreased relative valeric acid content in fecal specimens, while OE intake increased the fecal content and relative content of valeric acid (Figure 5A,C). Previous studies have reported that nicotinamide, a form of vitamin B, restored the reduction in valeric acid in the large intestinal contents of mice with DSS-induced chronic colitis [38]. Unlike other major SCFA, the physiological role of valeric acid in the colonic environment is poorly understood. Although limited studies are available, valeric acid reportedly has beneficial effects on the development of colitis by promoting intestinal epithelial cell proliferation [39] and inhibiting histone deacetylases [40]. Moreover, there was a strong correlation between the relative fecal content of valeric acid and the relative abundance of *Turicibacter* (Appendix A). A recent study has reported that, among *Turicibacter* species, *Turicibacter bilis* can produce valeric acid while *Turicibacter sanguinis* cannot [41]. In the present study, the 16S rRNA amplicon sequence could identify some species; for *Turicibacter*, identification was only possible up to the genus level. When OE was administered to an acute experimental colitis mouse model, no significant changes in fecal valeric acid content were observed [15]. However, the increase in fecal valeric acid content following OE intake was at least partially responsible for the anti-colitis effect, and *Turicibacter* may be involved in valeric acid production.

Patients with IBD [42] and experimental colitis mouse models [16] have shown reduced α-diversity in the gut microbiota community. The Chao-1 and Simpson indices indicate the estimate of the total community and a combination of richness and evenness, respectively [43]. DSS administration decreased the Simpson index, whereas OE intake tended to increase the Simpson index when compared with the control + DSS group (*p* = 0.06, Figure 6B). Furthermore, OE intake could slightly restore the structure of the fecal microbiota to that observed in normal mice (control group) (Figure 6C). Other food components, including radish sprout [44] and shrimp peptide [45], have also been reported to improve gut bacterial diversity reduced by DSS administration. Thus, OE intake may improve α- and β-diversity, exacerbated by DSS administration.

Moreover, an increase in the relative abundance of Verrucomicrobia and Proteobacteria has been reported in experimental colitis mouse models [46] and patients with IBD [47]. DSS administration enriched Proteobacteria (Figure 7) and increased their relative abundance (Figure 6D). However, the relative abundance of Proteobacteria in the OE + DSS group was unaltered when compared with that in the control + DSS group. Verrucomicrobia could not be detected in the 16S rRNA amplicon sequences employed in the present study. The OE + DSS group exhibited a lower Firmicutes/Bacteroidetes (F/B) ratio, which is used as an indicator of intestinal inflammation, owing to the higher relative abundance of Bacteroidetes than the control + DSS group (Figure 6E,F). This result is consistent with our previous study [15]. The F/B ratio was reportedly increased in a DSS-induced experimental colitis mouse model [48]; however, the F/B ratio was unaltered with or without DSS administration in the present study. However, whether the OE-mediated decrease in the F/B ratio suppresses the development of DSS-induced chronic experimental colitis remains unknown.

In the present study, the relative abundances of *Bifidobacterium*, *[Ruminococcus]* and *Ruminococcus* were reduced in chronic experimental colitis (Figure 6H–J). These alterations support data from previous studies examining dysbiosis in a DSS-induced experimental colitis mouse model [49,50]. It has been reported that the relative abundance of *Bifidobacterium*, which was decreased by DSS administration, increased in mice fed conjugated linoleic acid [51], but OE intake did not increase the relative abundance of *Bifidobacterium*. The relative abundance of *Ruminococcus* was higher in the OE + DSS group than in the control + DSS group (Figure 6J). In addition, there was a strong correlation between the relative abundance of *Ruminococcus* and the indicators of chronic experimental colitis (Appendix A). *Ruminococcus* is a Gram-positive bacterium that contributes significantly to butyric acid production in the colon by utilizing fiber and resistant starch [52]. However, OE intake did not alter the fecal content or relative content of butyric acid (Figure 5A,C). Conversely, *Enterococcus* can induce pathological processes in IBD [53], and *Enterococcus*-generated gelatinase can destroy the intestinal epithelium by activating protease-activated receptor 2 [54]. Moreover, *Enterococcus* richness is positively associated with IBD severity in young patients [55]. DSS administration increased the relative abundance of *Enterococcus* (Figure 6K), whereas OE intake tended to reduce the relative abundance of *Enterococcus* when compared with the control + DSS group (*p* = 0.07, Figure 6K). In this study, no evidence was obtained to show how *Ruminococcus* and *Enterococcus* are associated in the DSS-induced chronic experimental colitis mouse model. Of these results, the reduction of relative *Enterococcus* abundance is consistent with our previous study [15]. However, OE intake-mediated alterations in *Ruminococcus* and *Enterococcus* may play a crucial role in ameliorating chronic experimental colitis.

The active components of OE that exert anti-colitis effects remain elusive. Glycogen is a major component of OE (32.5 g/100 g). A previous study has shown that enzymatically synthesized glycogen functions like resistant starch [56]. We have observed that glycogen present in OE is resistant to digestive enzymes in a gastrointestinal digestion model (T. Ishida, H. Matsui, Y. Matsuda, R. Hosomi, K. Fukunaga, and M. Yoshida, unpublished data). It has been reported that glycogen intake markedly increased SCFA production and improved the microbiota in the cecum [56]. Therefore, glycogen, which is rich in OE, is thought to be the main component that exerted the anti-colitis effect of OE intake. In contrast, OE is rich in taurine (5.5 g/100 g), which can reportedly prevent DSS-induced colitis [57], and zinc (37.7 mg/100 g), which is reportedly associated with IBD [58]. We speculate that the anti-colitis effect mediated by OE intake may not be attributed to a single component but rather to the combined effects of several components.

## 5. Conclusions

Herein, we demonstrated that OE intake ameliorates the symptoms of chronic experimental colitis induced by repeated administration of DSS, in part by improving the gut microbiota community and fecal SCFA composition. Therefore, OE intake could be used in supplements and functional food materials to prevent and improve colitis. The observed results warrant further clinical studies to establish OE intake as a dietary strategy to improve the colonic environment.

## Figures and Tables

**Figure 1 foods-11-02032-f001:**
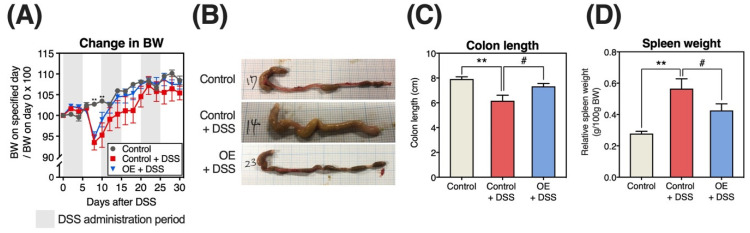
Indicators of chronic experimental colitis severity induced by dextran sulfate sodium (DSS). (**A**) Changes in body weight (BW). (**B**) Representative colon images. (**C**,**D**) Colon length and relative spleen weight in mice at day 30 after DSS administration. Results are presented as the mean ±  standard error of the mean (*n* = 6 for the control group and *n* = 8 for groups the control + DSS and oyster extract (OE) + DSS groups). ** *p* < 0.01 and # *p* < 0.05.

**Figure 2 foods-11-02032-f002:**
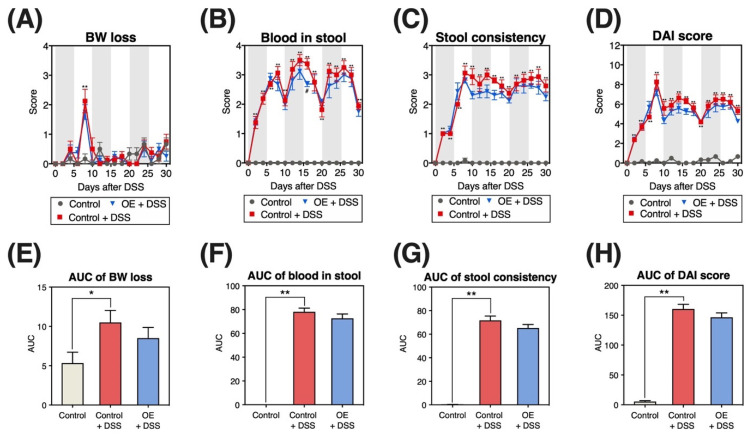
Disease activity index (DAI) scores. (**A**–**D**) Changes in body weight (BW) loss, blood in the stool, stool consistency, and DAI score during dextran sulfate sodium (DSS) administration. DAI score is the sum of BW loss, blood in the stool, and stool consistency scores. (**E**–**H**) Area under the curve (AUC) of BW loss, blood in the stool, stool consistency, and DAI score during DSS administration. Results are presented as the mean ±  standard error of the mean (*n* = 6 for the control group and *n* = 8 for groups the control + DSS and oyster extract (OE) + DSS groups). * *p* < 0.05, ** *p* < 0.01, and # *p* < 0.05.

**Figure 3 foods-11-02032-f003:**
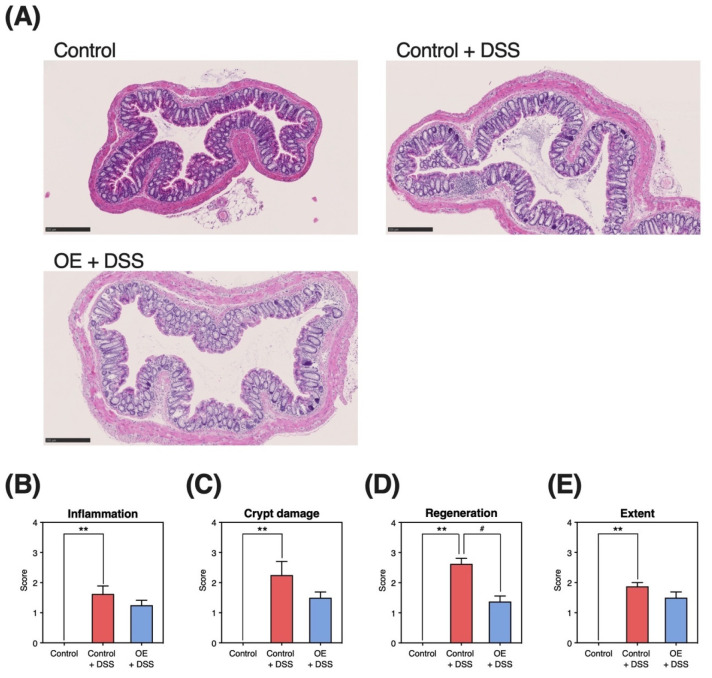
Histopathological grading of colonic tissue. (**A**) Representative histopathological sections (scale bar = 250 µm [10×]). (**B**–**E**) Histopathological grading scores of inflammation, crypt damage, regeneration, and extent in colonic tissues at day 30 after dextran sulfate sodium (DSS) administration. Results are presented as the mean  ±  standard error of the mean (*n* = 6 for the control group and *n* = 8 for groups the control + DSS and oyster extract (OE) + DSS groups). ** *p* < 0.01 and # *p* < 0.05.

**Figure 4 foods-11-02032-f004:**
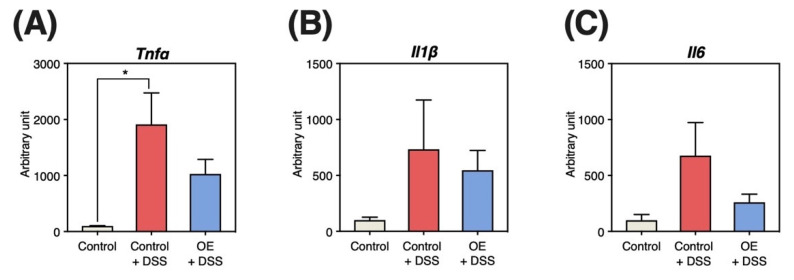
Expression levels of inflammation-related genes in colonic mucosa. (**A**–**C**) Expression levels of tumor necrosis factor (*Tnf) α*, interleukin (*Il) 1β*, and *Il6* in the colonic mucosa at day 30 after dextran sulfate sodium (DSS) administration. Results are presented as the mean  ±  standard error of the mean (*n* = 6 for the control group and *n* = 8 for groups the control + DSS and oyster extract (OE) + DSS groups). * *p* < 0.05.

**Figure 5 foods-11-02032-f005:**
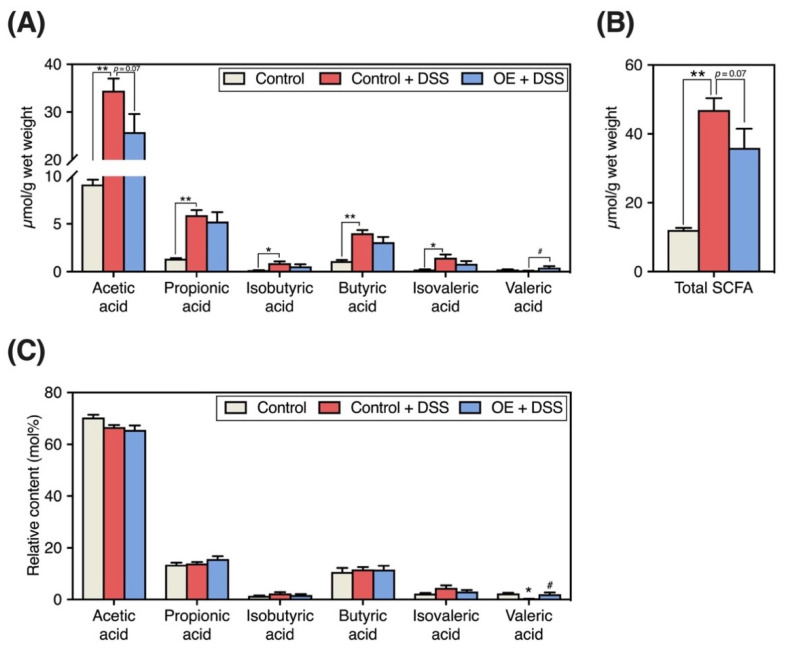
Fecal short-chain fatty acids (SCFA) compositions. (**A**,**B**) Fecal individual and total SCFA contents. (**C**) Fecal relative content of SCFA. Results are presented as the mean  ±  standard error of the mean (*n* = 6 for the control group and *n* = 8 for groups the control + dextran sulfate sodium (DSS) and oyster extract (OE) + DSS groups). * *p* < 0.05, ** *p* < 0.01, and # *p* < 0.05.

**Figure 6 foods-11-02032-f006:**
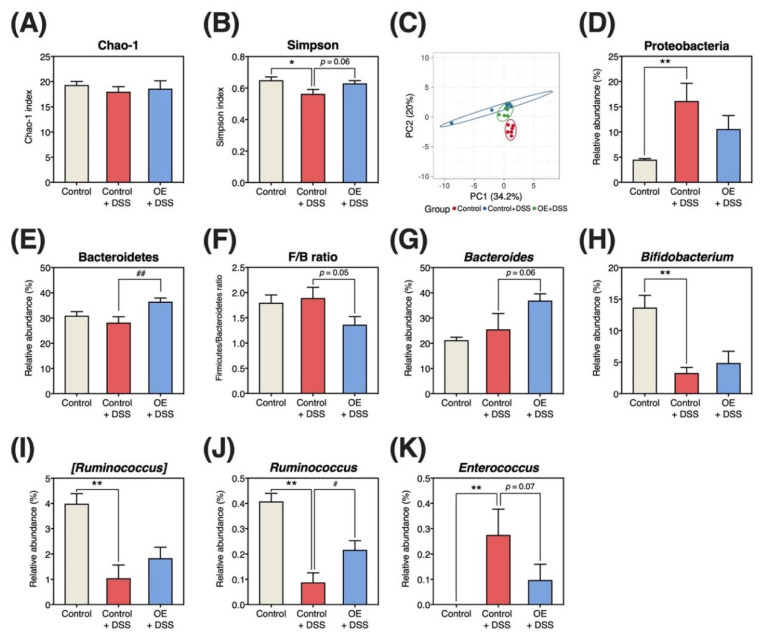
Microbiota community (structure and composition). (**A**) Chao-1 index. (**B**) Simpson index. (**C**) Principal component analysis of the bacterial genus community. (**D**,**E**,**G**–**K**) Relative abundance of each bacteria. (**F**) Firmicutes/Bacteroidetes (F/B) ratio. Results are presented as the mean  ±  standard error of the mean (*n* = 6 for all groups). * *p* < 0.05, ** *p* < 0.01, # *p* < 0.05, and ## *p* < 0.01. DSS, dextran sulfate sodium; OE, oyster extract.

**Figure 7 foods-11-02032-f007:**
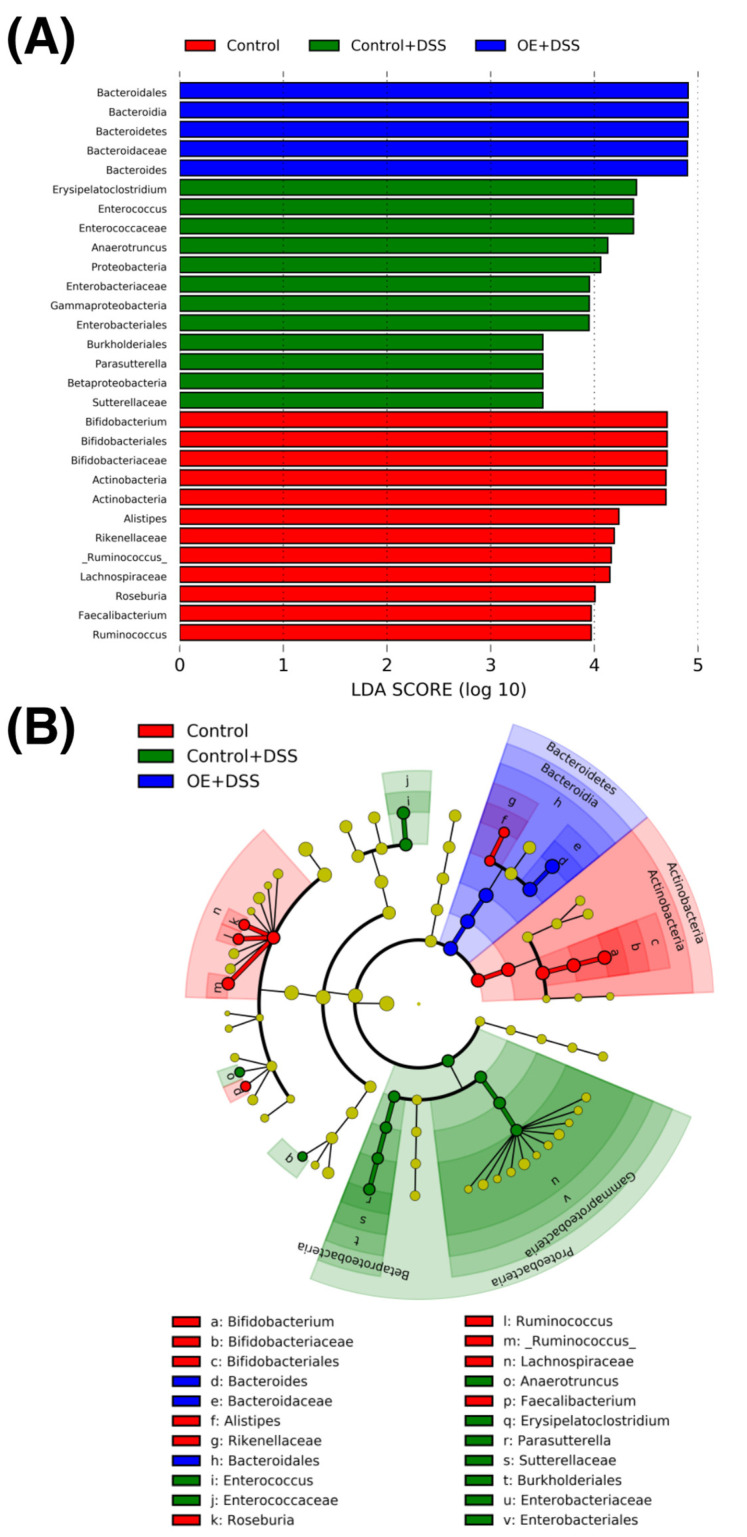
Comparisons of fecal microbiota using the linear discriminant analysis effect size analysis. (**A**) Results. (**B**) Cladogram. Parameters: linear discriminant analysis scores log_10_ > 3 and *p* < 0.05. DSS, dextran sulfate sodium; OE, oyster extract.

## Data Availability

Data associated with this article will be shared upon reasonable request with the corresponding author.

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
