# Peer review of "Dietary Oyster (Crassostrea gigas) Extract Ameliorates Dextran Sulfate Sodium-Induced Chronic Experimental Colitis by Improving the Composition of Gut Microbiota in Mice"

_foods, 2022, doi:10.3390/foods11142032_

Round 1

Reviewer 1 Report

This manuscript investigates how oyster extract intake ameliorates chronic experimental colitis induced by repeated DSS administration in mice, and the reason was to improve the gut environment, including the microbiota community and SCFA composition. The manuscript is interesting and well organized, but still needs some revision:

1. The abstract should be more informative by giving real results rather than elastic sentences. Important and main contents should be given. Support the results with some quantitative data.

2. It is better to describe more detail in methods, e.g., “2.6. Analysis of Gene Expression” and “2.7. Analysis of Fecal SCFA Compositions”.

3. Line 151 p-value Ë‚ 0.01 should be added.

4. Be sure to mention a significant level for comparing means, such as Lines 204, 230.

5. The authors should not simply described the main findings, the comparison of different results from those published papers especially the authors previous paper reference[15] should be analyzed and discussed.

6. Why oyster extract can improve microbiota community and SCFA composition? Which component in oyster extract may play a vital role in such effect? should be analyzed and discussed.

7. Conclusion: what is the future of your findings? Conclusion is not insightful, what are suggestions?

Reviewer 2 Report

The submitted manuscript treats an interesting question about the possibility of dietary modification of chronic inflammatory bowel disease (IBD) by oyster extract (OE).

Despite the clear presentation of the methods used and the results obtained, there are some areas were amendments should be made:

Lines 82-88 of the Materials and Methods. In the diet description, it should be described how the two diets, with and without OE, were balanced with respect to protein, carbohydrate and mineral contents.

Lines 107 of the Materials and Methods. It should be described how euthanasia was induces, by increased of anesthetic dose, by CO2, etc.

Lines 146-148 of the Materials and Methods. In the statistical procedures description, it should be explained why no evaluation was performed on the statistical significance of the differences between the control and OE+DSS groups. Even better, these differences could be evaluated and presented in the Results (even if a one-way ANOVA will be necessary).

Lines 271-end of the Discussion. The Discussion is focused mainly on bacterial flora changes and SCFA profile, while these seems to be non-essential and secondary to the DSS colitis development (e.g., doi:10.1155/2012/718617) and no proper discussion on the alleviation (or not) of morphological changes and inflammatory mediators by the OE. Also, no proper discussion is made about the regenerative response decrease by OE - is this good, or not so?

Also, some misspellings and style changes could improve the readability of the manuscript:

Line 22 of the Abstract. "(2) control + DSS (control diet + DDS), and (3) OE + DSS (OE diet + DDS)" - DDS is probably a misspelled DSS; otherwise, this abbreviation should be explained.

Line 75 of the Materials and Methods. "2.2. Approval for Animal Experimentation" - Experiments instead of Experimentation sounds better.

Lines 33, 39 and 43 of the Supplementary material. "position correlations" in the captions of figures S4, S5 and S6 are probably misspelled positive correlations?

Line 39 of the Supplementary material. "relative composition of each SCFA" in the caption of figure S5 is probably relative contents?

Line 46 of the Supplementary material. "relative fecal composition of valeric acid" in the caption of figure S7 and on the abscissa label of the same figure S7 is probably relative fecal contents?

Round 2

Reviewer 1 Report

The quality has been improved.